# A Trajectory Probability Network for City-Scale Road Volume Prediction

## Abstract

City-scale road volume prediction is a fundamental task in traffic management. However, the observation data are often incomplete and biased, posting a challenge for accurate prediction. Existing methods address this issue through interpolation techniques or manual priors, but they typically provide only a deterministic restoration, overlooking the influence of other potential scenarios. To overcome these limitations, we propose a novel neural network-based probabilistic model, the Trajectory Probability Network (TraPNet), which predicts traffic volume through the aggregation of the joint distribution of potential trajectories. TraPNet makes full use of current observations, historical data, and road network information to offer a comprehensive inference of road volumes. Unlike autoregressive methods, TraPNet makes predictions in a single step, substantially reducing computational time while maintaining high predictive accuracy. Experiments on real-world road networks demonstrate that TraPNet outperforms state-of-the-art methods, and can keep the advantage with only 20% observation ratio. The code will be made publicly available.

## 1 Introduction

Traffic volume prediction is a crucial task in urban traffic management, offering valuable insights into traffic congestion, road safety, and infrastructure planning. This task involves estimating the number of vehicles passing through each road at specific times, with predictions derived from current observations, historical data, and road network information.

Researchers have developed various methods to predict traffic volume, including traditional time series models Vlahogianni et al. (2014), deep learning models Lv et al. (2014), and graph neural networks Yu et al. (2017); Li et al. (2017). Many of these methods rely on historical in-route data collected via sensors deployed across road networks or GPS services Fang et al. (2020); Chen et al. (2024). However, gathering complete, city-wide traffic data remains a challenge: GPS data cannot capture all vehicles, and sensors are typically deployed only at key intersections, resulting in data that is both incomplete and unevenly distributed.

To address this challenge, some studies have focused on checkpoint-based data, providing a more accessible alternative Chen et al. (2023). Traditional methods often rely on prior probabilities to estimate missing traffic volumes Yu et al. (2023); Bao et al. (2023), while deep learning-based approaches reconstruct missing trajectories using current observations Zhang et al. (2019); Guo et al. (2024). However, the majority of these methods provide only deterministic reconstructions of the missing data, overlooking the inherent uncertainty in other potential scenarios. Furthermore, in some underdeveloped areas, observational data can be extremely sparse, rendering these methods less applicable.

To provide a more comprehensive prediction, we propose a novel probabilistic road volume prediction model. In our approach, we treat the volume on each road as the sum of random variables, with probabilities determined by individual vehicles on the road network. For each vehicle, we estimate the probability of its presence on any given road at each time step, rather than determining a single fixed trajectory. By accumulating these probabilities for all vehicles, we aggregate the distributions of all potential trajectories, leading to a more thorough prediction of road volumes. The posterior probability given incomplete observations is inferred using a neural network.

In addition to the integration mechanism, we enhance the model's robustness by fully utilizing different views of the data. Our model, Trajectory Probability Network (TraPNet), integrates current observations, historical trajectories, and road network information to provide a comprehensive inference of road volumes. All heterogeneous data are embedded into a unified latent space, and trajectory probabilities are estimated through a multi-view attention mechanism. Experiments on real-world road networks demonstrate that TraPNet outperforms state-of-the-art methods in both accuracy and efficiency. Notably, even when the observation ratio is as low as 20%, our model maintains its advantage. The primary contributions of this paper are as follows:

- We propose TraPNet, a neural network that leverages diverse sources of information, integrating the joint distribution of all potential trajectories to predict road volume.

- TraPNet performs complete volume prediction in a single step, significantly reducing computational time while maintaining high predictive accuracy.

- TraPNet demonstrates exceptional tolerance to missing data. With only 20% observations, TraPNet outperforms other models that require 50% observation ratio. This makes TraPNet applicable to a wider range of urban traffic scenarios.

## 2 RELATED WORK

Traffic volume prediction is a crucial task in urban traffic management, with various approaches developed over the years. These approaches can be broadly categorized into three types: traditional methods Vlahogianni et al. (2014), deep learning methods Ma et al. (2015); Yu et al. (2017), and checkpoint-based methods Chen et al. (2023). Traditional methods typically rely on historical in-route data collected by sensors or GPS services, utilizing models such as ARIMA. Deep learning methods, which use models like LSTM and GNN, also require complete volume data for accurate prediction. In contrast, checkpoint-based methods focus on incomplete data collected from key intersections, making them more applicable to real-world scenarios.

Despite their effectiveness, these methods face several challenges. Both traditional and deep learning approaches struggle with acquiring complete citywide data, limiting their practicality in real-world scenarios. Although checkpoint-based methods are more accessible, they also fail to handle extremely incomplete data scenarios. Furthermore, many existing methods overlook historical data or road network information, which can lead to inaccurate predictions.

### 2.1 TRAFFIC VOLUME PREDICTION

Traffic volume prediction has traditionally relied on historical in-route data, captured by sensors deployed across road networks or GPS service, using models like LSTMs and GNNs Zhang et al. (2017); Li et al. (2017); Yu et al. (2017); Diao et al. (2019). However, the challenge of acquiring complete, citywide data makes these methods impractical in real-world settings. To address this challenge, some studies focus on checkpoint-based data from key intersections, offering a more accessible alternative Liu et al. (2020); Kalander et al. (2020); Liu et al. (2018). While these methods predict traffic for regions or entire cities, they often lack the granularity to forecast at a city-wide per-road level. In addition, time series models, though effective at capturing temporal dependencies, suffer from cumulative errors and long inference times, limiting their scalability for large urban networks Ma et al. (2015); Zhao et al. (2019); Wong et al. (2022); Huang et al. (2023).

### 2.2 TRAJECTORY INTERPOLATION

In many real-world scenarios, incomplete trajectories are the primary source of data, necessitating the use of interpolation or reconstruction methods. One common approach relies on prior probabilities, assuming that missing trajectories follow specific patterns, such as the shortest route Patterson et al. (2020); Hunter et al. (2013); Iio et al. (2023). However, this approach can be inaccurate, as real vehicles often deviate from these assumptions. Another approach utilizes deep learning-based reconstruction, which, while effective, often relies solely on current observations without incorporating historical data or road network information. Additionally, both probabilistic and deep learning methods typically assume that vehicles follow a single fixed path, overlooking the possibility of

multiple potential routes Tang et al. (2023). Moreover, these methods remain inadequate in handling scenarios with extremely sparse data.

# 3 PROBLEM DEFINITION

## 3.1 BASIC NOTATIONS

Suppose the road network is represented as a graph $\mathbb{G} = (\mathbb{V}, \mathbb{E})$, where $\mathbb{V}$ is the set of nodes and $\mathbb{E}$ is the set of edges. $\mathbb{V} = \{1, 2, ..., V\}$, which are the number of intersections (node). $\mathbb{E} = \{e_1, e_2, ..., e_E\}$, $e_i = (o_i, d_i, l_i)$, which represents the origin, destination and weight of the $i_{th}$ road (edge). The weight can be length, average speed, traffic level or any other information of the road.

Considering that the number of nodes is much smaller than the number of edges, we use nodes to represent trajectories. A trajectory can be represented as a sequence of nodes, $[v_1, v_2, ..., v_T]$, where $v_i$ is a rearrangement of $\{0, 1, ..., V\}$, $T$ is the maximum time step, 0 means that the car disappears. To indicate the time spent on each road, we repeat $v_i$ in the trajectory, e.g. $[1, 2, 2, 2, 2, 3, 4, 0]$ means that a car spend 1 time step on edge (1,2), 4 steps on (2,3), 1 step on (3,4), and finally disappear. We denote all the trajectories as $X \in \{0, 1, ..., V\}^{B,T}$, where $B$ is the number of observed trajectories.

## 3.2 ROAD-LEVEL VOLUME PREDICTION

In real-world scenarios, traffic is recorded by a limited number of unevenly distributed sensors, such as cameras at intersections, resulting in incomplete observations. For example, assume that only node 1 and node 3 are observable, in the former case where the complete trajectory is $[1, 2, 2, 2, 2, 3, 4, 0]$, the corresponding observation should be $[1, 0, 0, 0, 0, 3, 0, 0]$. Given all the observed trajectories during T time steps, Road-Level Volume Prediction is to estimate the volume of each road at each time, where the volume is defined as the number of vehicles passing through a given road. The volume is denoted as $\mathbf{Vol} \in \mathbb{R}^{E,T}$, where $\mathbf{Vol}[i, t]$ is the volume of road $i$ at time $t$.

Trajectory Probability is the distribution of $X$. We assume that each trajectory is independently distributed, and denote the trajectory probability as $Y \in [0, 1]^{B,T,V}$, where $Y[b, t, v]$ is the probability that the $b_{th}$ vehicle at time $t$ is on $v$. Once the trajectory probability distribution is obtained, the volume can be either calculated by the expectation or the MAP estimation. In this paper, we focus on predicting volume by the aggregation of $q_\theta(Y|X, X_{his}, A)$, where $\theta$ is the parameter of our model, $X_{his}$ is the historical trajectory information and $A$ is the road network information.

# 4 METHOD

In this section, we introduce our model, Trajectory Probability Network (TraPNet). First, we introduce the overall architecture of the model in section 4.1. The process of embedding different types of information is described in section 4.2. The core component of our model, the Multi-view Attention Block, is detailed in section 4.3. Finally, we discuss the model's optimization and the aggregation to inference road volume in section 4.4.

## 4.1 OVERALL ARCHITECTURE

The overall architecture of our model is shown in fig. 1. The model consists of three components: embedding layers, multi-view attention blocks, and linear projections. The embedding layers project the observed trajectories, road network, and historical trajectories into the same hidden space. These different sources of embeddings are then aggregated within the multi-view attention block, which captures complex relationships across different views. The output from the multi-view attention block is passed through a linear projection layer with softmax activation to compute the trajectory probabilities. Finally, the trajectory probabilities are used to estimate the road volume.

## 4.2 EMBEDDING LAYERS

As shown in fig. 2, the embedding layers project all the real-world information into the aligned token space. The embedding layers consist of three parts: observation embedding, history embedding

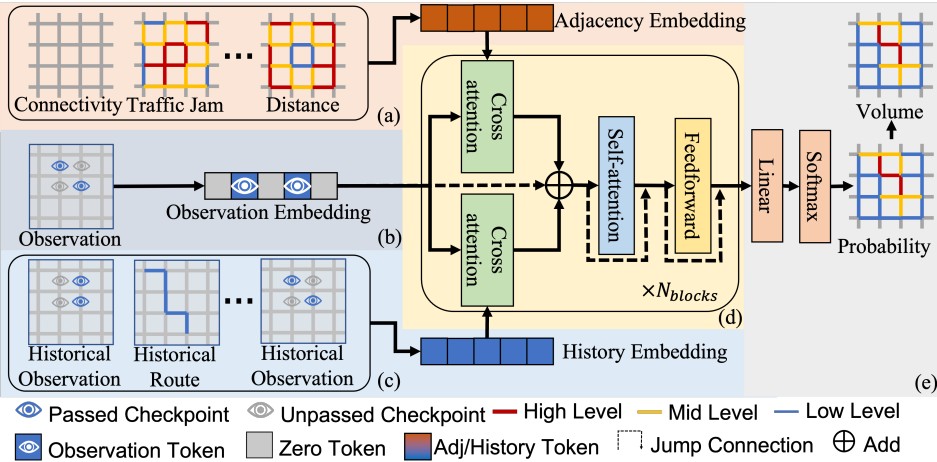

Figure 1: Overview of the proposed TraPNet. (a, b, c) represent the embedding layers for road network data, observation data, and historical data, respectively. (d) denotes the multi-view attention block. (e) corresponds to the output projection.

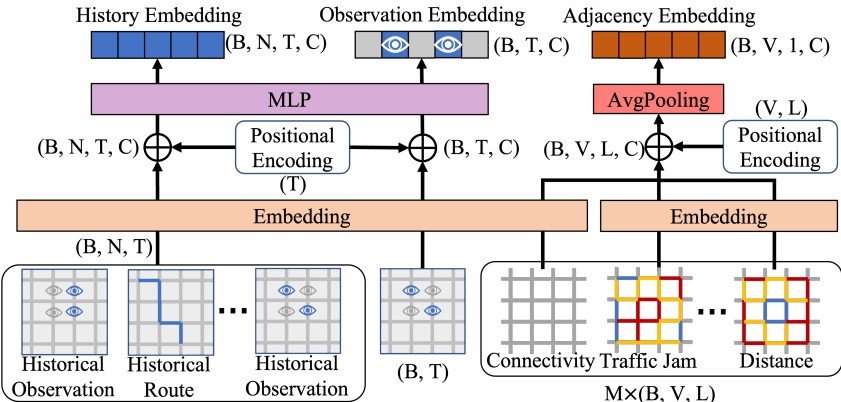

Figure 2: Embedding layers of TraPNet.

and road network embedding. The observation embedding and history embedding share the same weights, while the road network needs other embedding tables.

Given a batch of observed trajectories $X \in \{0, 1, ..., V\}^{B,T}$, we first project the discrete node into continuous tokens by an embedding matrix $E_{traj} \in \mathbb{R}^{V+1,C}$, where $C$ is the dimension of the latent space. Combined with the positional encoding over the time steps, the tokens are further passed through an MLP layer to get the final observation tokens. The MLP layer consists of a linear projection, a layer normalization, and a SiLU activation function. The output of the MLP layer is denoted as $z_{obs} \in \mathbb{R}^{B,T,C}$. To be specific, the observation embedding is calculated as follows:

$$z_{obs} = \text{SiLU}(\text{LN}(\text{Linear}(E_{traj}(X) + E_{pos}^T))) \tag{1}$$

where LN denotes LayerNorm. $E_{pos}^T \in \mathbb{R}^{B,T,C}$ is the positional encoding over T.

The historical data consist of past trajectories, which may be either complete or incomplete. Incomplete trajectories refer to past observations of the target vehicle recorded at checkpoints. Complete trajectories, on the other hand, can be obtained from sources such as GPS data, past trajectory probability estimations, or other relevant datasets. If no historical data are available, they are set to zeros. Since historical data are also in the form of trajectories, they are processed similarly to the observation data. The history embedding is computed as follows:

$$z_{his} = \text{SiLU}(\text{LN}(\text{Linear}(E_{traj}(X_{his}) + N \times E_{pos}^T))) \tag{2}$$

where $X_{his} \in \{0, 1, ..., V\}^{B,N,T}$ is the historical data, $N$ is the number of historical trajectories. $E_{pos}^T$ is repeat $N$ times to match the dimension.

The road network data are represented as adjacency tables, denoted as $A = [A_0, A_1, ..., A_M]$, where $M$ is the number of adjacency tables. $A_0 \in \{0, 1, ..., V\}^{B,V,L}$ is a special adjacency table that represents the road network connections, where $L$ is the max connectivity. $A_0[b, v, l]$ is the $l_{th}$ neighbor of node $v$ for the $b_{th}$ trajectory, and 0 indicates no additional neighbors. The other adjacency tables provide information about roads, such as speed limits, distances, or road types, and are aligned with $A_0$ to ensure that $A_i[b, v, l]$ corresponds to the same road as $A_0[b, v, l]$. If $A_i$ contains continuous data, it can be projected into tokens by MLP layers; if $A_i$ contains discrete data, an embedding matrix is used. To reduce computation, the continuous data could be discretized into several bins, and an embedding matrix is then applied. The road network embedding is calculated as follows:

$$z_{adj} = \text{AvgPooling}(E_{traj}(A_0) + \sum_{i=1}^{M} E_{adj}(A_i) + E_{pos}^{V,L})$$ (3)

where $A_i \in \{1, ..., K\}^{V,L}$ is the discretized road weights, $K$ is the discretization level, $E_{adj} \in \mathbb{R}^{K,C}$ is the embedding matrix of the adjacency tables, $E_{pos}^{V,L} \in \mathbb{R}^{B,V,L,C}$ is the positional encoding over (V, L). We use average pooling to reduce the computation load.

### 4.3 MULTI-VIEW ATTENTION BLOCK

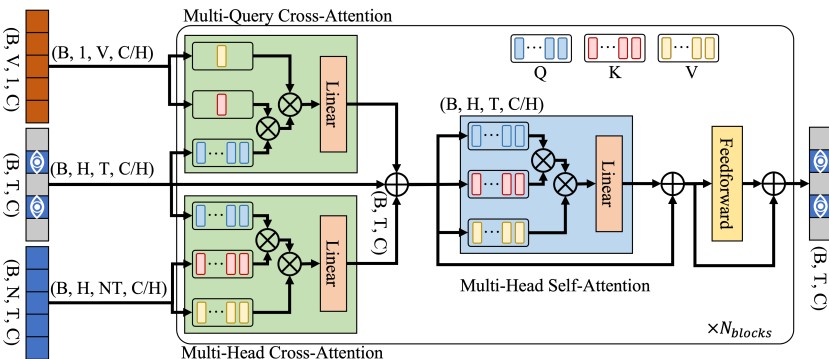

Figure 3: Multi-view attention block of TraPNet.

The multi-view attention block is the core component of our model, which integrates the observation information, history information, and road network information. As shown in fig. 3, the multi-view attention block consists of two cross-attention blocks and one self-attention block, each wrapped with residual connections and LayerNorm. We adopt multi-head attention and multi-query attention Ainslie et al. (2023) mechanism to capture the complex relationship between different views. The adjacency tokens represent the information of each node, which is comprehensive but not efficient. To address this problem, we adopt the multi-query attention mechanism to reduce the computation. The adjacency tokens are linearly projected into the key and value tokens, while the observation tokens are linearly projected into the query tokens. As demonstrated in our ablation study, the multi-query attention mechanism effectively reduces computation with minimal impact on model performance. The multi-view attention block is computed as follows:

$$z_{adj}' = \text{MQA}(z_{obs}, z_{adj}, z_{adj})$$
$$z_{his}' = \text{MHA}(z_{obs}, z_{his}, z_{his})$$
$$z_{obs}' = z_{obs} + z_{adj}' + z_{his}'$$ (4)
$$z_{obs}' = \text{MQA}(z_{obs}', z_{obs}', z_{obs}') + z_{obs}'$$
$$z_{obs}'' = \text{FFN}(z_{obs}') + z_{obs}'$$

where MQA and MHA denote the multi-query attention and multi-head attention, which project the inputs into query, key, and value tokens accordingly. FFN denotes the feed-forward network, which

consists of two linear projections and a SiLU activation function. $z'_{adj}, z'_{his}, z'_{obs}$ are intermediate tokens and $z''_{obs}$ is the output of the multi-view attention block.

## 4.4 OPTIMIZATION AND AGGREGATION

After the multi-view attention block, the hidden tokens are projected into the trajectory probability by a linear projection and a softmax activation function. The output is calculated as follows:

$$q_\theta(Y|X, X_{hist}, A) = \text{Softmax}(\text{Linear}(z''_{obs})) \tag{5}$$

where $\theta$ is the parameter of our model, $Y \in [0,1]^{B,T,V}$ is the estimated trajectory probability. Given the ground truth trajectory probability $p(Y)$, we optimize the KL-divergence between $p(Y)$ and $q_\theta(Y|X, X_{hist}, A)$ to train our model. The loss function is calculated as follows:

$$\mathcal{L} = \sum_{b=1}^{B} \sum_{t=1}^{T} \sum_{v=1}^{V} p(Y[b,t,v]) \log \frac{p(Y[b,t,v])}{q_\theta(Y[b,t,v]|X[b], X_{hist}[b], A[b])} \tag{6}$$

where $p(Y[b,t,v])$ and $q_\theta(Y[b,t,v]|X[b], X_{hist}[b], A[b])$ are both Bernoulli distributed.

We train TraPNet using complete trajectories. Given a road network, we randomly assign checkpoints with a ratio $\alpha$. The complete trajectories are masked according to the checkpoints. When $N + 1$ trajectories of the same vehicle are available, we randomly select one trajectory as the observation and use the remaining as historical data. Since the real trajectory is complete, the probability degenerates into a one-point distribution: $p(Y[b,t,v]) = 1$ if $v = X[b,t]$, otherwise $p(Y[b,t,v]) = 0$. Consequently, the loss function simplifies to the cross-entropy loss. The final loss function is calculated as follows:

$$\mathcal{L} = -\sum_{b=1}^{B} \sum_{t=1}^{T} \sum_{v=1}^{V} \tilde{Y}[b,t,v] \log q_\theta(Y[b,t,v]|X[b], X_{hist}[b], A[b]) \tag{7}$$

where $\tilde{Y}[b,t,v]$ is one-hot encoding of the real trajectory. Details of the training process are shown in section 5.1.1.

The road volume is aggregated by the expectation of the trajectory probabilities. First, the node-represented trajectory probability is transformed into edge-represented trajectory probability by the multiplication of the origin and destination node probabilities. To be specific:

$$\dot{Y}[b,t,i] = Y[b,t,o_i] \times Y[b,t+1,d_i] + Y[b,t,o_i] \times Y[b,t+1,o_i] \tag{8}$$

where $\dot{Y}[b,t,i]$ is the probability that the $b_{th}$ car is on edge $e_i$ at time $t$. This edge probability consists of two components: (1) the probability that the vehicle moves to edge $e_i$, and (2) the probability that the vehicle remains on edge $e_i$. The term $Y[b,t,o_i] \times Y[b,t+1,d_i]$ corresponds to the first situation, representing the probability that the vehicle moves from node $o_i$ to node $d_i$. The term $Y[b,t,o_i] \times Y[b,t+1,o_i]$ represents the second situation, where the vehicle stays on edge $e_i$. Since the volume is the number of cars that pass through the road, it can be estimated by summing the expected contributions of each vehicle. To be specific:

$$\mathbf{Vol}[i,t] = \sum_{b=1}^{B} \left( \frac{\dot{Y}[b,t,i]}{\sum_{j=0}^{E} \dot{Y}[b,t,j]} \right) \tag{9}$$

where $\mathbf{Vol}[i,t]$ is the volume of road $i$ at time $t$. We apply normalization to make sure that the total volume contribution from each car sums to 1.

## 5 EXPERIMENTS

In this section, we comprehensively evaluate the proposed method on real-world road networks. We begin by describing the experiment setup in section 5.1, including the data preparing and hyperparameters. Next, we present the main results in section 5.2, offering both an overall comparison and a detailed analysis of performance. Finally, we conduct an ablation study in section 5.3 to assess the impact of different computation-efficient mechanisms.

## 5.1 Experiment Settings

### 5.1.1 Data Preparing

The experiment is conducted on two real-world cities: Boston and Jinan. The Boston road network consists of 241 nodes and 369 edges. We randomly select the origin and destination of each vehicle, assign random weights to the roads, and simulate the trajectories using the shortest path. The maximum time step is set to 60. For each pair of origin and destination, we simulate 5 trajectories, meaning that each observed trajectory is accompanied by 4 historical trajectories. In total, we simulate 500,000 trajectories for training and 10,000 trajectories for testing. The weighted adjacency matrix for each simulation is recorded as the road network input.

The Jinan road network data is obtained from Yu et al. (2023), consisting of 8,908 nodes and 23,312 edges. In addition to position and connectivity information, this dataset includes road length, road type, and complete trajectories of 963,125 individual vehicles. We randomly select 800,000 trajectories for training, with the remaining trajectories used for testing. For each vehicle, we randomly select 1 trajectory as the observation and 4 trajectories as historical inputs. When necessary, we apply repeatable sampling to obtain the 4 historical trajectories. We use road length as the road weight for the road network inputs. The time scales of the trajectories vary significantly, ranging from seconds to hours. To standardize the data, we rescale the time scope to 60 time steps.

### 5.1.2 Parameters and Computational Setting

Table 1: Default hyperparameters.

| Dataset | Model Parameters | | | | Training Parameters | | | |
| --- | --- | --- | --- | --- | --- | --- | --- | --- |
| | Blocks | Hidden size | Heads | FFN expansion factor | Discretization factor | Batch size | Lr | Epochs |
| Boston | 8 | 64 | 16 | 2 | w/o | 512 | 0.01 | 20 |
| Jinan | 8 | 64 | 16 | 2 | 30 | 50 | 0.01 | 100 |

The experiments are conducted on a server equipped with 4 NVIDIA A30 GPUs. We use Stochastic Gradient Descent (SGD) as the optimizer, along with a Cosine Annealing learning rate scheduler. The default hyperparameters are presented in table 1. For the Boston road network, being relatively small, we utilize an MLP as the tokenizer rather than employing the discretization mechanism. For the larger Jinan road network, we apply the discretization mechanism to reduce computational overhead.Since most trajectories do not reach the maximum of 60 steps, we apply a mask over the loss function to ignore the padding steps. The masked value for the loss between the prediction and padding steps is set to 0.0001, ensuring that the training primarily focuses on real steps while allowing the output trajectories to terminate at the padding steps. The training for the Boston dataset takes approximately 8 GPU hours, while training on the Jinan dataset takes about 100 GPU hours. Each training process is repeated 3 times, and we report the average results. During each training iteration, the checkpoints are randomly selected with the default ratio $\alpha = 0.5$.

## 5.2 Main Results

In this section, we present the main results of our model on the Boston and Jinan road networks. We compare TraPNet with two state-of-the-art methods: Cam-Traj-Rec Yu et al. (2022) and Traj2Traj Liao et al. (2023). Cam-Traj-Rec assigns prior distributions to the missing trajectories based on road weights and infers the posterior distribution given the observed trajectories. Traj2Traj is a deep learning method for trajectory reconstruction. We provide the overall Mean Absolute Error (MAE) comparison in section 5.2.1 and visualize the road volume predictions in sections 5.2.2 and 5.2.3.

### 5.2.1 Overall MAE Comparison

The overall MAE comparison is presented in table 2 and fig. 4, where we show the MAE of different methods under varying checkpoint ratios. TraPNet consistently achieves lower MAE compared to the other two methods. When the checkpoint ratio is 0.1, the MAE of TraPNet is approximately

Table 2: Overall MAE comparisons under different checkpoint ratios.

| | **Boston** | | | | | **Jinan** | | | | |
|---|---|---|---|---|---|---|---|---|---|---|
| Checkpoint ratio | 0.1 | 0.2 | 0.3 | 0.5 | Time | 0.1 | 0.2 | 0.3 | 0.5 | Time |
| Cam-Traj-Rec | 7.29 | 4.08 | 3.33 | 1.63 | 93.2 s | 0.355 | 0.300 | 0.248 | 0.179 | 83.7 s |
| Traj2Traj | 4.71 | 3.15 | 2.56 | 1.25 | 41.5 s | 0.297 | 0.249 | 0.232 | 0.143 | 57.6 s |
| TraPNet | 2.24 | 1.22 | 1.07 | 0.667 | 3.35 s | 0.214 | 0.125 | 0.126 | 0.071 | 18.1 s |

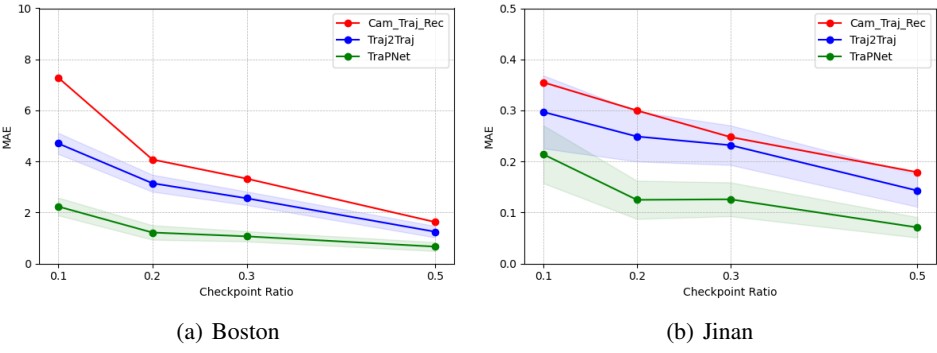

(a) Boston  (b) Jinan

Figure 4: Overall MAE comparison under different checkpoint ratios.

20% lower than that of the other methods. With 10% observation ratio, TraPNet can get the similar performance as Cam-Traj-Rec with 50% observation ratio. With 20% observation ratio, TraPNet already outperforms other methods with 50% observation ratio. In addition, TraPNet is significantly faster than the other two methods, and on Boston it can achieve almost real-time performance.

Cam-Traj-Rec is a prior-based method that assigns prior distributions to the missing trajectories based on road weights. The prior distribution is calculated according to the length of different routes between the origin and destination. As an interpolation-based method, Cam-Traj-Rec struggles to handle missing trajectory segments at the beginning and end, making its MAE highly sensitive to the checkpoint ratio, which affects the length of the missing parts.

Traj2Traj is an LSTM-based trajectory reconstruction method that can address missing segments at both the beginning and end of trajectories. However, as an autoregressive method, Traj2Traj suffers from error accumulation as the time step increases. Additionally, since it relies solely on current observations and the road network, its predictions become less reliable when the observation data are highly incomplete.

### 5.2.2 VOLUME PER ROAD COMPARISON

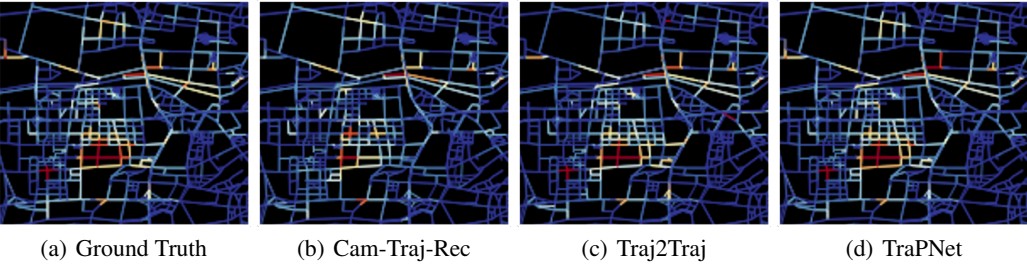

(a) Ground Truth  (b) Cam-Traj-Rec  (c) Traj2Traj  (d) TraPNet

Figure 5: Volume per road comparison. Blue means low volume and Red means high volume. We focus on the downtown areas, the visualization of the whole city can be found in appendix A

As shown in fig. 5(b), in the central part of the downtown area, a major issue of Cam-Traj-Rec is that the predicted volume is biased by the prior distribution. This results in misjudgments regarding

which roads are the busiest. In contrast, the deep learning-based method Traj2Traj provides predictions that are closer to the ground truth, but they still lack accuracy. As seen in the top-left part of fig. 5(c), while Traj2Traj correctly captures the relationships between roads, the absolute volume predictions are not precise. For a more detailed view, please refer to the supplementary material, where we provide videos showing the volume distribution across roads at each time step.

### 5.2.3 VOLUME PER TIME STEP COMPARISON

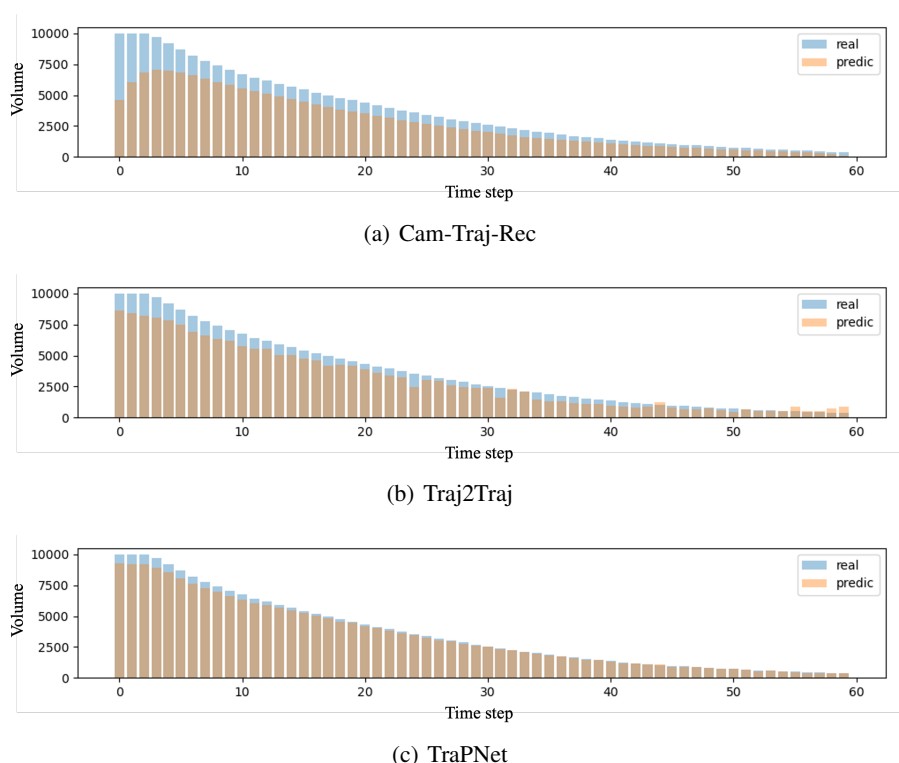

(a) Cam-Traj-Rec

(b) Traj2Traj

(c) TraPNet

Figure 6: Volume per time step comparison. Blue bars are the ground truth volumes of all the roads and orange bars are predictions.

We visualize the volume per time step in fig. 6. As shown in fig. 6(a), Cam-Traj-Rec performs poorly at the beginning. This is a common limitation of interpolation-based methods, which struggle to handle missing data at the beginning and end of a trajectory. In contrast, fig. 6(b) demonstrates that Traj2Traj performs better than Cam-Traj-Rec, but its performance becomes unstable as the time step increases. A possible explanation is that LSTM-based methods are sensitive to long-term dependencies. Additionally, autoregressive methods face challenges in determining when a trajectory should end, especially when the input is highly incomplete. According to fig. 6(c), TraPNet's performance is both stable and accurate. The predictions closely match the ground truth, and early-stage errors do not adversely influence later predictions. There are, however, some minor inaccuracies at the beginning, likely because the softmax function makes it difficult for the model to confidently predict a value of 1.

### 5.3 ABLATION STUDY

We conduct an ablation study to assess the impact of various computation-efficient mechanisms. These experiments are limited to the Boston dataset, as the "BVLC" token shape is too large for the Jinan dataset. The results are summarized in table 3.

From lines (2) and (3), we observe that introducing both historical data and road network information significantly reduces the MAE, with the road network information playing a more crucial role. Lines (4) and (5) demonstrate that the discretization mechanism effectively reduces computation

Table 3: Ablation Study on Boston.

| | Data hyperparameters | | | Structure hyperparameters | | Performance | |
|---|---|---|---|---|---|---|---|
| Index | History | Adjacency | Discretization | Adj embedding shape | Adj attention type | MAE | Time |
| (1) | w/ | w/ | w/o | BV1C | Multi-query | 0.667 | 3.35 s |
| (2) | w/o | w/ | w/o | BV1C | Multi-query | 1.12 | 3.01 s |
| (3) | w/ | w/o | - | - | - | 2.21 | 1.13 s |
| (4) | w/ | w/ | 10 | BV1C | Multi-query | 0.716 | 2.22 s |
| (5) | w/ | w/ | 20 | BV1C | Multi-query | 0.683 | 2.41 s |
| (6) | w/ | w/ | w/o | BVLC | Multi-query | 0.451 | 12.7 s |
| (7) | w/ | w/ | w/o | B11C | Multi-query | 0.894 | 1.77 s |
| (8) | w/ | w/ | w/o | BV1C | Multi-head | 0.538 | 9.50 s |

without significantly impacting performance. Additionally, lines (6) and (7) show that the token shape of the adjacency table has little effect on performance but significantly affects time complexity. Finally, line (8) indicates that the multi-query attention mechanism also reduces computation while maintaining performance stability.

# 6 DISCUSSION

## 6.1 BALANCE BETWEEN PERFORMANCE AND EFFICIENCY

TraPNet is designed to achieve a balance between performance and computational efficiency. The multi-view attention mechanism effectively integrates observational data, historical information, and road network information, enhancing performance but also introducing a substantial computational burden. The discretization mechanism and multi-query attention mechanism mitigate this burden by slightly compromising performance. Similarly, the pooling mechanism strikes a balance between efficiency and GPU memory usage. On smaller road networks, these mechanisms may be optional, depending on specific requirements, but for larger road networks, they are essential for practical deployment.

## 6.2 THE CHOICE OF ONE-HOT LABELS

Our primary goal is to estimate trajectory probabilities given incomplete observations. However, during the training process, we rely exclusively on complete trajectories as the ground truth. An alternative approach would be to use prior distributions as the ground truth, which can be derived from road weights. However, as demonstrated in section 5, manually assigned prior distributions can introduce bias. To ensure accuracy and reliability, we use only complete trajectories as the ground truth. In the future, we aim to explore the potential of adapting manual prior distributions to more closely reflect real-world distributions.

# 7 CONCLUSION

In this paper, we proposed a novel probabilistic approach to address the challenge of city-scale road volume prediction with incomplete observations. We introduced Trajectory Probability Network (TraPNet), a model capable of estimating trajectory probabilities based on incomplete observations, historical trajectories, and road network information. The road volume can be comprehensively estimated by aggregating the trajectory probabilities. TraPNet's one-step road volume prediction, combined with various computation-efficient mechanisms, ensures both high performance and computational efficiency. We conducted extensive experiments on real-world road networks, demonstrating that TraPNet outperforms state-of-the-art methods. With only 20% observations, TraPNet outperforms other models that require 50% observation ratio. Furthermore, our ablation study highlights the impact of different computation-efficient mechanisms. TraPNet is highly adaptable to real-world scenarios, as its inputs can be flexibly defined.

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

## A   FULL VISUALIZATION OF THE VOLUME DISTRIBUTION

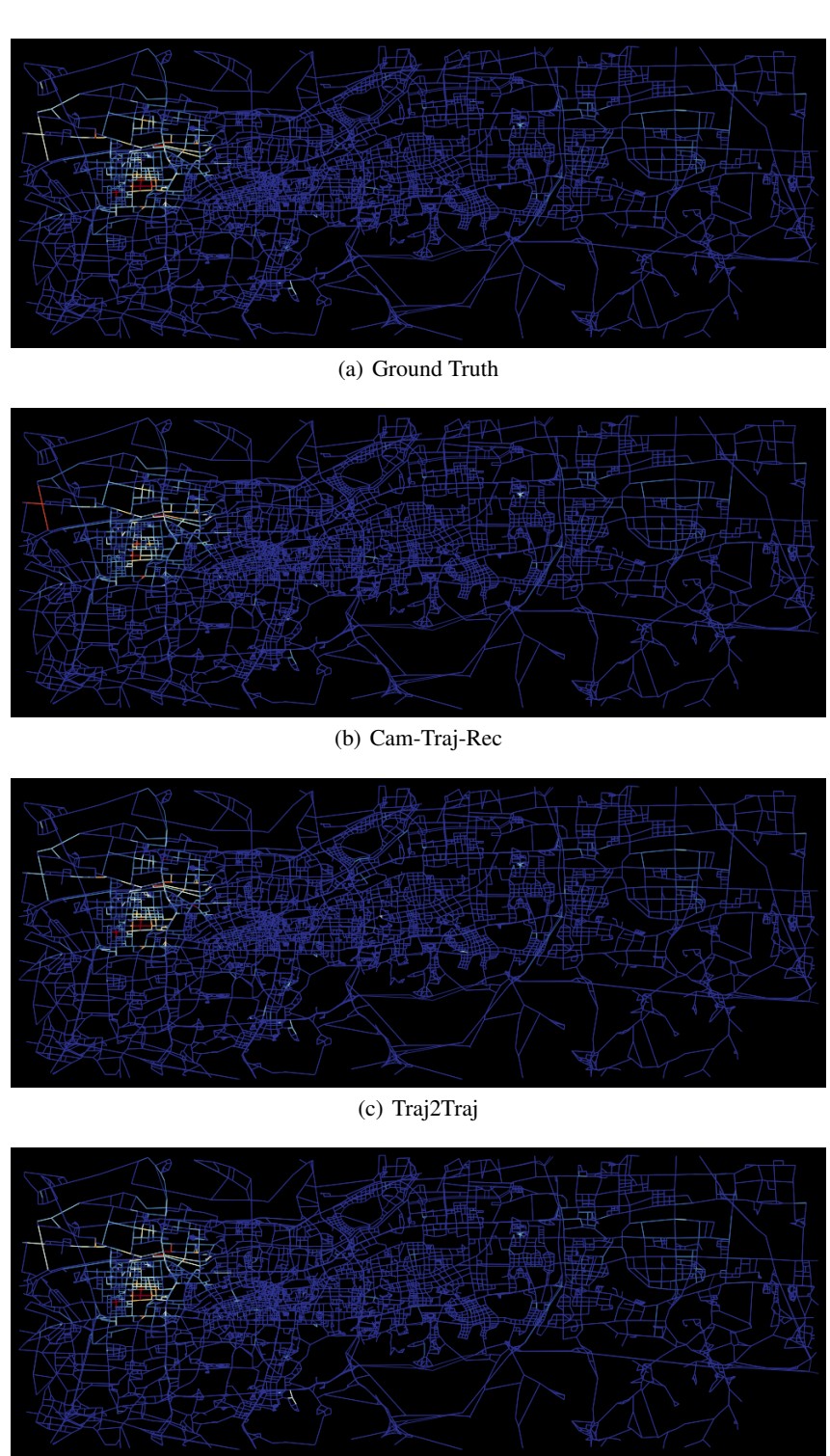

(a) Ground Truth

(b) Cam-Traj-Rec

(c) Traj2Traj

(d) TraPNet

Figure 7: Volume per road comparison. Blue means low volume and Red means high volume.

