# OpenReview forum: "A Trajectory Probability Network for City-Scale Road Volume Prediction"
_ICLR.cc/2025/Conference — Submitted to ICLR 2025_

### Official Review · Reviewer_P8Ra · 2024-10-30

**Soundness:** 1
**Presentation:** 1
**Contribution:** 2
**Rating:** 3
**Confidence:** 4

**Summary:**

The proposed microscopic approach to traffic volume prediction is both intriguing and innovative. By breaking down aggregated traffic volume into individual vehicle movements, this method has the potential to achieve superior accuracy and expand its applications beyond mere traffic volume prediction.

**Strengths:**

The proposed microscopic approach to traffic volume prediction is both intriguing and innovative. By breaking down aggregated traffic volume into individual vehicle movements, this method has the potential to achieve superior accuracy and expand its applications beyond mere traffic volume prediction.

**Weaknesses:**

1. The framing of the problem—traffic volume prediction—appears very similar to traffic flow prediction, which is a prominent topic in spatiotemporal data mining and intelligent transportation systems. Given this context, it seems unusual that the proposed method is not compared with any of the numerous existing traffic flow prediction methods.
2. The presentation of challenges could be improved. The paper states that "the majority of these methods provide only deterministic reconstructions of missing data, overlooking inherent uncertainty in other potential scenarios." The term "inherent uncertainty" is somewhat vague and lacks further elaboration. Additionally, since MAE is used as the main evaluation metric in experiments, the probabilistic prediction capability of the proposed method isn't thoroughly assessed. This raises questions about whether the method can truly account for uncertainty in traffic comprehensively.
3. Another challenge mentioned is that "in some underdeveloped areas, observational data can be extremely sparse." However, it remains unclear how this challenge is addressed by the proposed method.
4. It seems that predicting future traffic volumes requires first simulating each vehicle's appearance probability in the city, raising concerns about computational efficiency—an aspect not assessed in experiments.
5. Some implementation details are missing. For instance, the calculation of adjacency tables $A$ is not clearly defined, leaving ambiguity regarding how "the other adjacency tables" are calculated and aligned with $A_0$.

**Questions:**

1. (Related to W1) Could the authors explain their rationale for not comparing their method with any existing traffic flow or spatiotemporal graph prediction methods?
2. (Related to W2 and W3) Could the authors elaborate on both challenges presented and explain how their proposed method explicitly addresses them?
3. (Related to W4) Could the authors provide justification regarding the seemingly high computational resources required by their proposed method?

---

### Official Review · Reviewer_EoGu · 2024-10-31

**Soundness:** 2
**Presentation:** 1
**Contribution:** 2
**Rating:** 3
**Confidence:** 4

**Summary:**

The paper proposes a probabilistic model that aggregates the joint distribution of potential trajectories for road-based traffic volume prediction. Leveraging multi-view attention blocks, the model integrates current observations, historical trajectories, and road network data to improve both predictive accuracy and efficiency. The model shows good performance on the Boston and Jinan road networks compared to two baseline models, even with high levels of missing data.

**Strengths:**

S1: The network employs a probabilistic approach to address the intrinsic uncertainty in trajectory data.

S2: The model demonstrates tolerance to missing data, making it suitable for applications in scenarios with sparse observations.

**Weaknesses:**

W1: The paper is hard to follow due to an unclear problem definition. It is not clear whether the focus is on predicting future traffic volumes or on trajectory recovery. In Section 3.2, it is also unclear whether T for Y represents past or future time steps. Additionally, the baselines address trajectory recovery, while the related works in Section 2.1 focus on future prediction.

W2: The descriptions of deep learning methods and checkpoint-based methods are confusing. Checkpoint-based methods appear to be a subset of deep learning methods, with the main difference from existing works lying in the data collection approach. However, it is unclear why checkpoint-based methods are considered “more applicable to real-world scenarios.”

W3: The trajectory representation may involve a high level of repetition. For example, repeating node 2 four times to indicate that the car spends four time steps on (2, 3) could result in substantial computational overhead and memory usage.

W4: Only MAE is reported as a performance metric. Additional metrics like RMSE and MAPE would provide a more comprehensive evaluation.

W5: The transition from Equation 6 to Equation 7 in the loss function is not clearly explained, especially regarding how the simplification occurs.

W6: The ablation study shows that using multi-head attention (8) yields an MAE of 0.538, significantly outperforming multi-query attention with an MAE of 0.667. Similarly, configuration (6) with the BVLC embedding shape achieves an MAE of 0.451, notably better than the default setting (0.667), contradicting the paper’s claim that the default setting “has little effect on performance.”

**Questions:**

Q1: Could the problem definition be clarified further?

Q2: What is the relationship between deep learning methods and checkpoint-based methods? What unique advantages do checkpoint-based methods have over existing deep learning methods?

Q3: How does the model achieve single-step predictions while reducing computational time and maintaining high accuracy? What mechanism drives this advantage?

Q4: The model estimates road volume by aggregating trajectory probabilities. What theoretical or practical justification supports this approach?

Q5: While multi-view attention effectively integrates diverse data sources, it may incur high computational and memory costs. Is there any comparison of training time, inference time, or memory usage?

Q6: What is the rationale behind using Multi-Query Attention (MQA) for adjacency tokens and observation data, but Multi-Head Attention (MHA) for historical data?

Q7: Are there any illustrations or results showing the confidence in the trajectory predictions?

---

### Official Review · Reviewer_5UVU · 2024-11-02

**Soundness:** 2
**Presentation:** 2
**Contribution:** 2
**Rating:** 3
**Confidence:** 4

**Summary:**

This paper introduces a framework named TraPNet that uses current observations, historical data, and road networks to predict road volume. While previous methods struggle with missing data, TraPNet demonstrates exceptional tolerance. Experimental results show that TraPNet outperforms other baselines.

**Strengths:**

1. The task road volume prediction is meaningful.
2. The modeling of current observations, historical data, and road networks is well-rounded. The design of TraPNet demonstrates a thoughtful balance between performance and computational efficiency, which is somewhat innovative.

**Weaknesses:**

1. There is notable redundancy in the paper, particularly in the related work section (e.g., the first two paragraphs and Section 2.1). Reducing this duplication could improve readability.
2. The  descriptions of trajectory interpolation/generation and traffic volume prediction are somewhat unclear. When the authors talk about predicting in a single step, they seem to be talking about autoregressive trajectory generation, while the task they focus on is traffic volume prediction.
3. Following 2, the relevance of the baselines, Cam-Traj-Rec and Traj2Traj, to the traffic volume prediction task is unclear, as these primarily address trajectory interpolation/generation. In addition, when considering trajectory generation, more advanced baselines need to be compared, e.g., MobilityGPT[1], and TS-TrajGen[2], which also show similar trajectory maps like Figure 5 in the paper.
4. The experimental dataset is insufficient. The data on Boston are all generated shortest path data, which may not fully capture real-world conditions. Given TraPNet’s use of cross-attention in the node dimension, I am concerned about the performance and efficiency of the model on larger road networks (10k, or 100k+ nodes).

[1] Haydari, Ammar, et al. "Mobilitygpt: Enhanced human mobility modeling with a gpt model."
[2] Jiang, Wenjun, et al. "Continuous trajectory generation based on two-stage GAN." Proceedings of the AAAI Conference on Artificial Intelligence.

**Questions:**

See weaknesses.

---

### Official Review · Reviewer_VsnL · 2024-11-04

**Soundness:** 2
**Presentation:** 3
**Contribution:** 2
**Rating:** 5
**Confidence:** 3

**Summary:**

The paper proposed a method to do accurate city-scale traffic prediction by full using of current observations, historical data, and road network information with a designed frameworks. The method is reasonable and the results outperforms some baselines.

**Strengths:**

1. The paper is easy to follow and the method sounds reasonable.
2. The authors compare the method with some baselines and  the results are good.

**Weaknesses:**

1. Missing intuitions and experiments. I think more experiments and validation should be added to explain why the current design outperform SOTA baselines by a large margin. It is difficult to understand based on current explanations and experiments, which only have some numbers, but not concrete evidence of why.
2. More most recent baselines should be added to validate the effectiveness.

**Questions:**

See weakness above

---

### Meta-Review · Area_Chair_DBBL · 2024-12-20

**Metareview:**

This paper presents a probabilistic model for traffic volume prediction through aggregate probabilities of future trajectories.
The model has a promise to be able to deal with missing data.
However, the paper needs a lot more work. The details of the methods and the justifications of the method design are missing. The paper lacks solid experiments and comparisons with state-of-the-art baselines. It also only uses one dataset.
All reviewers agreed that this paper is not yet ready for publication.

**Additional Comments On Reviewer Discussion:**

There is no further discussion, as there was no response from authors during rebuttal. The reviewers are also unanimous in their recommendation.

---

### Decision · Program_Chairs · 2025-01-22

Reject